# Targeted Next-Generation Sequencing Identifies Additional Mutations Other than *BCR∷ABL* in Chronic Myeloid Leukemia Patients: A Chinese Monocentric Retrospective Study

**DOI:** 10.3390/cancers14235752

**Published:** 2022-11-23

**Authors:** Shiwei Hu, Dan Chen, Xiaofei Xu, Lan Zhang, Shengjie Wang, Keyi Jin, Yan Zheng, Xiaoqiong Zhu, Jie Jin, Jian Huang

**Affiliations:** 1Department of Hematology, The Fourth Affiliated Hospital of Zhejiang University School of Medicine, N1 Shangcheng Road, Yiwu 322000, China; 2Zhejiang Provincial Clinical Research Center for Hematological Disorders, Hangzhou 310003, China; 3Department of Hematology, The First Affiliated Hospital of Zhejiang University School of Medicine, No. 79 Qingchun Road, Hangzhou 310003, China

**Keywords:** chronic myeloid leukemia (CML), next-generation sequencing (NGS), molecular response (MR), tyrosine kinase inhibitor (TKI), scoring system

## Abstract

**Simple Summary:**

TKI have vastly improved long-term outcomes for patients with CML, although it is still hard for a proportion of patients to obtain ideal molecular responses. Advances in NGS technology have enabled the incorporation of somatic mutation profiles in classification and prognostication. With an increased focus on achieving deep molecular responses, we try to explore the risk conferred by additional genomic lesions other than *BCR∷ABL* through NGS technology. We also figure out how clinical characteristics, distinct TKI options and risk scores influence the achieving of molecular responses. This research has the potential to lay the foundation for improved risk classification according to clinical and genomic risk and to enable more precise early identification of TKI.

**Abstract:**

A proportion of patients with somatic variants show resistance or intolerance to TKI therapy, indicating additional mutations other than *BCR∷ABL1* may lead to TKI treatment failure or disease progression. We retrospectively evaluated 151 CML patients receiving TKI therapy and performed next-generation sequencing (NGS) analysis of 22 CML patients at diagnosis to explore the mutation spectrum other than *BCR∷ABL1* affecting the achievement of molecular responses. The most frequently mutated gene was *ASXL1* (40.9%). *NOTCH3* and *RELN* mutations were only carried by subjects failing to achieve a major molecular response (MMR) at 12 months. The distribution frequency of *ASXL1* mutations was higher in the group that did not achieve MR^4.0^ at 36 months (*p* = 0.023). The achievement of MR^4.5^ at 12 months was adversely impacted by the presence of >2 gene mutations (*p* = 0.024). In the analysis of clinical characteristics, hemoglobin concentration (HB) and MMR were independent factors for deep molecular response (DMR), and initial 2GTKI therapy was better than 1GTKI in the achievement of molecular response. For the scoring system, we found the ELTS score was the best for predicting the efficacy of TKI therapy and the Socal score was the best for predicting mutations other than *BCR∷ABL*.

## 1. Introduction

Chronic myeloid leukemia (CML) is a myeloproliferative neoplasm characterized by reciprocal translocation between chromosomes 9 and 22 (t (9;22) (q34.1; q11.2)) and the formation of the *BCR∷ABL1* fusion gene on the Philadelphia chromosome [1]. The *BCR∷ABL1* fusion gene encodes tyrosine kinase, leading to a chronic phase of CML manifested by clonal expansion of leukemic cells and indolent symptoms. The discovery of tyrosine kinase inhibitors (TKIs) has led to long-term disease control and has drastically revolutionized the prognosis of CML patients. The average life expectancy of CML patients treated with TKI is near that of the general population patients [2]. However, therapy still fails in a proportion of patients.

Although *BCR∷ABL1* mutations remain the major mechanisms of TKI resistance [3], *BCR∷ABL1*-independent mechanisms contributing to TKI resistance are poorly understood [4]. Questions remain on how to predict treatment failure and how to select frontline TKI therapy at the time of diagnosis. At present, there are no routine testing strategies to predict the molecular response of TKI therapy, but advances in next-generation sequencing (NGS) may aid in expanding genomic analysis in the management of CML patients. The increasingly mainstream use of NGS represents a sensitive and resource-efficient alternative for genetic research. It has been documented that NGS plays a vital role in the stratification of prognosis and evaluation of therapeutic effects in patients with acute myeloid leukemia (AML), whereas few studies concentrate on clinically relevant variants in CML patients, especially on variants in addition to *BCR∷ABL1* kinase domain mutations. Recently, researchers have found a variety of somatic mutations in addition to those in *BCR∷ABL1* in myeloid malignancies and indicate that additional mutations could contribute to disease progression. CML patients with poor outcomes carried mutated genes such as *ASXL1*, *IKZF1*, *RUNX1*, *DNMT3A,* and *CREBBP* at diagnosis more frequently [5,6]. This technology has great potential in revealing additional genetic events to recognize patients with poor therapeutic response.

In addition to genetic events, clinically relevant baseline data can also influence the molecular response to TKIs. Second-generation TKIs (2GTKI), nilotinib and dasatinib, provide CML patients with more options for first- or second-line CML therapy. A recent study showed that there were no significant differences in efficacy and safety between original and generic imatinib treatment [7]. However, with a total of at least four available TKI options, there is still a challenge when choosing the optimal first-line TKI to achieve the best therapeutic response and deep molecular response (DMR). National Comprehensive Cancer Net (NCCN) guidelines recommend using the Sokal, Hasford, EUTOS and ELTS scoring systems for CML-chronic phase (CML-CP) patients prior to the initiation of TKI therapy [8,9,10,11,12]. Although risk scores have been developed to predict the responses and/or outcomes of CML patients, few studies have critically compared them as predictors in the evaluation of molecular responses.

In this study, we performed NGS analysis on 161 candidate mutations to explore the mutation spectrum in addition to those in *BCR∷ABL1* influencing the response to TKI treatment and prognosis. We also compared clinical and hematological characteristics in 151 consecutive subjects with CML-CP treated by 1GTKI or 2GTKI and validated four scoring systems in the prediction of TKI efficacy.

## 2. Materials and Methods

### 2.1. Subjects

We conducted a retrospective study of 151 CML-CP patients in the Fourth Affiliated Hospital of Zhejiang University School of Medicine from October 2014 to December 2020. Patients with incomplete information from laboratory tests and medical records were excluded. Data of covariates determined at diagnosis included sex, age, WBC and platelet counts, hemoglobin concentration (HB), percentage of EOS and BAS and spleen size. Sokal, Hasford, EUTOS and ELTS scores at diagnosis were calculated as previously described [8,9,10,11]. Therapy responses and outcomes were extracted from medical records or obtained by follow-up. The study was approved by the Ethics Committee of the Fourth Affiliated Hospital of Zhejiang University School of Medicine and conducted in accordance with the Declaration of Helsinki.

### 2.2. NGS Detection and Response Assessment

We performed NGS analysis of 22 CML patients at diagnosis. Genomic DNA was purified from bone marrow or peripheral blood with a Gentra Puregene Blood Kit (Qiagen, Hilden, Westphalia, Germany) according to the manufacturer’s protocol. High-throughput gene sequencing was performed using ultrahigh multiple PCR exon enrichment technology with an average sequencing depth of 800×. Mutation analysis was performed using the Ion Reporter System (ThermoFisher Scientific, Waltham, Massachusetts, The United States) and Variant Reporter Software (ThermoFisher Scientific, Waltham, Massachusetts, The United States).

Response definitions were as follows: (1) early molecular response (EMR): *BCR∷ABL*^IS^ ≤ 10% at 3 months; (2) *BCR∷ABL*^IS^ ≤ 1% at 6 months; (3) major molecular response (MMR): *BCR∷ABL*^IS^ ≤ 0.1% at 12 months; (4) molecular response 4.0 (MR^4.0^): *BCR∷ABL*^IS^ ≤ 0.01%; (5) molecular response 4.5 (MR^4.5^): *BCR∷ABL*^IS^ ≤ 0.0032%; and (6) deep molecular response (DMR) involving MR^4.0^ and MR^4.5^. Progression-free survival (PFS) was calculated from TKI start to progression (accelerated phase or blast phase), death or censored at last follow-up.

### 2.3. Statistical Analysis

Statistical analyzes were performed using SPSS statistics 26.0 (International Business Machines Corporation, Armonk, State of New York, The United States)and GraphPad Prism 8.0 software (GraphPad Software, San Diego, California, The United States). Categorical covariates were reported as percentages and counts. Continuous variables were reported as medians and ranges. For comparisons among these groups, the Pearson chi-square, continuity correction and Fisher’s exact test were used for categorical factors, single factor analysis of variance or t test was used for normally distributed continuous variables, and the Kruskal–Wallis test (3 groups) or Mann–Whitney U test (2 groups) was used for continuous variables that did not conform to the normal distribution. The association between the clinical characteristics or molecular characteristics and PFS was calculated using the Kaplan–Meier method with the log-rank test. *p* < 0.05 was considered as statistically significant.

## 3. Results

### 3.1. Mutation Analysis Based on NGS Detection

A total of 151 CML patients were included in the study. The median follow-up by the data cut-off was 73 months (range, 12–234 months). There were 83 men and 68 women with a median age at presentation of 45 years (range, 18–92 years). Baseline characteristics of the total CML-CP patients and the subgroup performed by NGS are displayed in Table 1. Mutation screening was performed by NGS in 161 hematologic malignancy-related variants of DNA samples from 22 subjects. In total, 25 genes and 51 mutations were detected, most of which were nonsynonymous SNVs (Figure 1A. The coexistence pattern among high-frequency variants was quite intricate (Figure 1B). Further analysis of the 25 genes with an allele mutation frequency (VAF) ≥ 5% revealed that *ASXL1* was the most frequently mutated gene (9/22, 40.9%), followed by *KMT2C* (6/22, 27.3%), *DIS3* (4/22, 18.2%), *ATM* (3/22, 13.6%), *DNMT3A* (3/22, 13.2%) and *NOTCH3* (3/22, 13.6%) (Figure 1C).

### 3.2. Mutation Analysis in the 12 M-MMR Group and Non12 M-MMR Group

In this cohort, 11 subjects achieved MMR at 12 months, and 11 subjects failed. We analyzed the mutation spectrum in these two groups and found differences in the distribution of mutated genes (Figure 2A,B). Although there was no significant difference between these two groups, *NOTCH3* (0% vs. 13.6%, *p* = 0.214) and *RELN* (0% vs. 9.1%, *p* = 0.476) mutations were only carried by subjects who failed to achieve MMR at 12 months, suggesting CML patients with *NOTCH3* and *RELN* mutations might have poor long-term treatment effects (Table 2).

### 3.3. Mutation Analysis in the 36 M-MR^4.0^ Group and Non-36 M-MR^4.0^ Group

We further divided the 22 subjects into another two groups according to the achievement of MR^4.0^ at 36 months. Sixteen subjects achieved MR^4.0^ at 36 months, and six subjects failed. The presence of mutations in these genes did not have any significant association with achievement of MR^4.0^ at 36 months, except in the case of *ASXL1* (25% vs. 83.3%, *p* = 0.023), suggesting that *ASXL1* mutation was an adverse factor for the achievement of MR^4.0^ (Table 3).

### 3.4. Analysis of the Number of Variants

The number of mutated genes also influenced the efficacy of TKI therapy. We regrouped the 22 subjects into two groups according to the achievement of MR^4.5^ at 24 months. Eleven subjects achieved failure, and 11 subjects failed at 24 months. The median number of mutated genes in subjects achieving and failing to achieve MR^4.5^ at 24 months was 1 (range, 0–3) and 4 (range, 0–6), respectively (*p* = 0.033) (Figure 3A). In the group that failed to achieve MR^4.5^ at 24 months, there were more subjects carrying more than two mutated genes (9.1% vs. 63.6%, *p* = 0.024), implying that it is less likely to achieve MR^4.5^ with the increase in the number of mutated genes and that the existence of more than two mutations is a poor prognostic factor for achieving DMR (Figure 3B).

### 3.5. Mutation Analysis of PFS

Finally, we chose six mutated genes with mutation frequencies greater than 10% and generated Kaplan–Meier curves to analyze the impact of mutational status at diagnosis on PFS. However, no statistical significance was found in the effects of mutations in *ASXL1* (*p* = 0.371), *KMT2C* (*p* = 0.079), *DIS3* (*p* = 0.467), *ATM* (*p* = 0.280), DNMT3A (*p* = 0.479) and *NOTCH3* (*p* = 0.479) on PFS in this cohort (Figure 4A–F).

### 3.6. Analysis of Clinical Characteristics on Molecular Response

Among 151 CML patients, 125 subjects reached MR^4.5^ at a median of 31 months (range, 2–179 months), and 26 subjects did not reach MR^4.5^ until the end of follow-up, of which 10 subjects had disease progression and 3 subjects died. Univariate analysis in this cohort found that age (*p* = 0.018), HB (*p* = 0.001) and *BCR∷ABL*^IS^ level at 3 months (*p* = 0.002), 6 months (*p* = 0.036) and 12 months (*p* < 0.001) were significantly correlated with the achievement of MR^4.5^ (Table 4). Multivariate analysis identified HB (relative risk, [RR], 1.023; *p* = 0.08) and *BCR∷ABL*^IS^ level at 12 months (RR, 2.485; *p* < 0.001) as independent predictive covariates for MR^4.5^ (Figure 5). Subjects reaching MMR at 12 months and with higher HB were more likely to reach MR^4.5^, whereas sex, other hematological indices and the four scoring systems had no statistical significance in predicting whether MR^4.5^ could be reached.

### 3.7. Analysis of TKI Therapies on Molecular Response

There were 142 subjects treated with 1GTKI, and 9 subjects received 2GTKI as first-line treatment (Group C). Among the 142 subjects treated with 1GTKI in the first line, 115 (81.0%) subjects continued receiving 1GTKI (Group A), and 27 subjects (17.9%) switched to 2GTKI (Group B). Among the 115 subjects who used 1GTKI continuously, 68 subjects chose original 1GTKI (Group A1, Glevic, Novartis, Basel, Switzerland), and 47 subjects chose generic 1GTKI (Group A2, Genike, Chiatai Tianqing Pharmaceutical Group Co., Ltd., Tianjin, China). The comparison results are shown in Table 5. We found significant differences between Group C and Group B in achieving EMR at 3 months (*p* = 0.012), *BCR∷ABL*^IS^ ≤ 1% at 6 months (*p* = 0.012) and MMR at 12 months (*p* = 0.018). The results suggested that subjects receiving 2 GTKI for the initial therapy benefited more in achieving molecular remission within a year. In the comparison between the two subgroups, we found that although Group A1 found it much easier to achieve EMR at 3 months (*p* = 0.025) than Group A2, there was no statistical significance in achieving *BCR∷ABL*^IS^ ≤ 1% at 6 months, MMR at 12 months and MR^4.5^ between them.

### 3.8. Analysis of Scoring Systems on Molecular Response and Mutation Status

The 151 subjects were further regrouped based on the Sokal, Hasford, EUTOS and ELTS scoring systems at diagnosis. We found that subjects stratified by the ELTS scoring system had significant differences in achieving EMR at 3 months (*p* = 0.001) and MMR at 12 months (*p* = 0.004) (Table 6). Subjects stratified by the EUTOS scoring system had a significant difference in achieving MMR at 12 months (*p* = 0.049) (Table 6). However, there was no significant difference in subjects stratified by the Hasford and Sokal scoring systems (Table 6). As such, the ELTS scoring system has an evident advantage in predicting molecular remission and the efficacy of TKI therapy compared with the other three scoring systems. We further analyzed the efficacy of four scoring system based on mutation status. We found Sokal score was statistically significant in distinguishing between “no mutations” and either “mutations” or “other mutations” (*p* = 0.001, *p* = 0.012) (Table 7). However, no statistical significance between “ASXL1 mutations” and either “no mutations” or “other mutations” was found in the four scoring systems.

## 4. Discussion

In this study, we performed NGS analysis on 161 candidate variants from 22 CML patients, demonstrating that gene mutations in addition to *BCR∷ABL1* were present in a significant proportion of patients. *ASXL1* was the most frequently mutated gene, and subjects with this mutation were less likely to achieve MR^4.0^ at 36 months, suggesting a reduced sensitivity to TKIs in CML patients with *ASXL1*. These conclusions were consistent with the latest studies showing that *ASXL1* mutations were the most common genetic lesions in CP at diagnosis and may confer a poor prognosis [4,13,14,15]. *ASXL1* mutations with VAF ≥ 17% were even related to poor responses to third-generation TKI therapy [16]. Mechanisms might be attributed to the characteristic of *ASXL1* being latent, initiating mutations that accumulate during the progression of CML, and the protein encoded by *ASXL1* has a mutual effect with *BCR∷ABL1* [17,18].

It was also found that mutations in *NOTCH3* and *RELN* were present only in subjects who did not achieve MMR at 12 months, indicating that CML-CP patients with these two mutations might have a poor response to TKI therapy. *NOTCH3* mutation may regulate the transcription of pTa and the activity of the NF-kB signaling pathway to promote tumor progression [19]. *RELN* mutation plays a role by enhancing glycolysis and activating the Akt/STAT3 pathway [20,21]. The detection of specific gene mutation mutations may assist in stratifying patients more accurately, providing information for prognosis and laying the basis for treatment strategies. We also investigated the relationship between the number of mutations and the efficacy of treatment. The results showed that the presence of more than two mutations was an adverse factor for achieving DMR, which may be related to the involvement of more than one signaling pathway and thus lead to the failure of treatment [14,22].

Next, we analyzed the correlations between clinical features and molecular response, drawing the conclusion that the HB value at diagnosis and *BCR∷ABL*^IS^ level at 12 months were two independent factors for MR^4.5^, which was consistent with the conclusions of several studies [23,24,25]. Although the HB value was not included in the four scoring systems, CML-CP patients with moderate anemia showed more aggressive characteristics, such as higher WBC counts and a higher percent of myeloblasts and BAS, than nonmoderate anemia patients [26]. This could be partly explained by the high levels of hematopoietic stem cells, which alter the components in the bone marrow microenvironment and elicit defective hematopoiesis in CML patients [27].

2GTKI could reduce the level of *BCR∷ABL*^IS^ more deeply and rapidly and lower the risk of progression to an accelerated phase or blast crisis [28,29,30,31]. We found that the administration of 2GTKI in the first line resulted in easier achievement of EMR at 3 months, *BCR∷ABL*^IS^ ≤ 1% at 6 months and MMR at 12 months, suggesting that the application of 2GTKI in the first line might benefit patients more in achieving earlier and higher response rates. In addition, there was no difference observed in long-term efficacy between original and generic 1GTKI, indicating that generic 1GTKI might be an attractive alternative for CML-CP patients due to its lower price and similar molecular remission compared with original 1GTKI, which was in accordance with the study conducted by Jiang H [7].

Among the four scoring systems, our study showed that risk stratification by the ELTS score had a high predictive value in treatment responses. Therefore, it was reasonable to point out that the ELTS scoring system was the most sensitive discriminator of TKI efficacy compared with other risk scores, followed by the EUTOS score. Although the EUTOS and ELTS scores were able to predict the MMR within 12 months, only the ELTS score could predict DMR at any time [32]. The ELTS score was also a better outcome predictor in addition to its advantage in predicting *BCR∷ABL*^IS^ levels, especially in subjects receiving initial 2GTKI therapy [33]. As for the ability to evaluate mutations, although the Socal score could well distinguish mutated subjects and non-mutated subjects, there was no ideal scoring system in predicting the mutation status, especially *ASXL1* mutations. Furthermore, this may lead to the inadequacy of the scoring system’s efficacy in predicting molecular response. This indicated *ASXL1* could serve as an additional prognostic factor and be incorporated into scoring systems to better predict the molecular response of CML patients.

## 5. Conclusions

In summary, we found that the *ASXL1* mutation and the presence of more than two mutations were adverse factors in the response to TKI treatment. The HB value and the achievement of MMR at 12 months were independent factors for DMR, and the initial 2GTKI therapy was better than 1GTKI for EMR and MMR. For scoring systems, we found that the ELTS score was the best in predicting the efficacy of TKI therapy and Socal score was the best in predicting mutations other than *BCR∷ABL*. Future genomic analysis may combine genomic data with clinical parameters to improve CML classification and prognostication. These results provide evidence and a basis for risk stratification and individualized treatment for CML-CP patients and warrant further investigation at a larger population level.

## Figures and Tables

**Figure 1 cancers-14-05752-f001:**
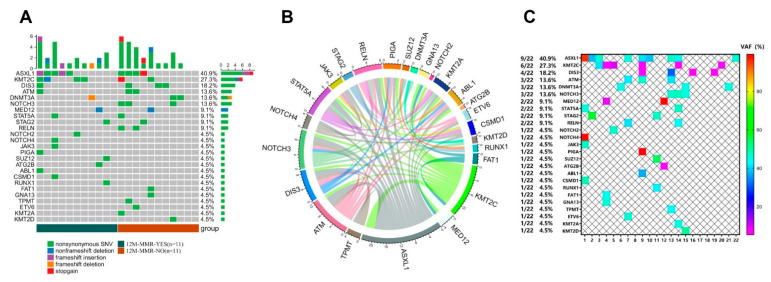
Mutation spectrum in CML-CP patients. (**A**) Type of mutation; (**B**) Co-occurrence of common variants; (**C**) VAF distribution.

**Figure 2 cancers-14-05752-f002:**
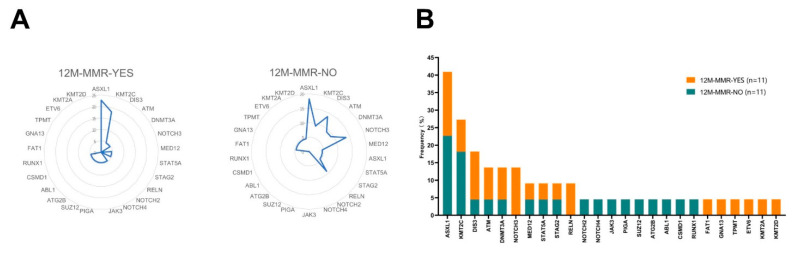
Mutation spectrum and frequency in the 12M-MMR group and non-12M-MMR group. (**A**) Mutation spectrum in the 12M-MMR group and non-12M-MMR group; (**B**) Mutation frequency in the 12M-MMR group and non-12M-MMR group.

**Figure 3 cancers-14-05752-f003:**
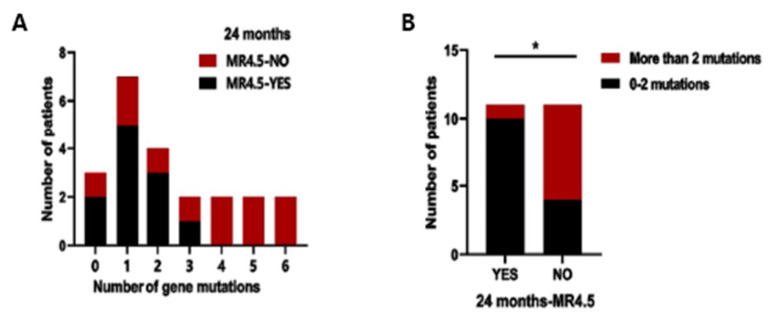
Proportion of mutations based on number of mutations. (**A**) Analysis of the average number of mutations based on gene function classification; (**B**) Proportion of mutations in the 24M-MR^4.5^ group and the non-24M-MR^4.5^ group based on the number of mutations. * *p*-value < 0.05.

**Figure 4 cancers-14-05752-f004:**
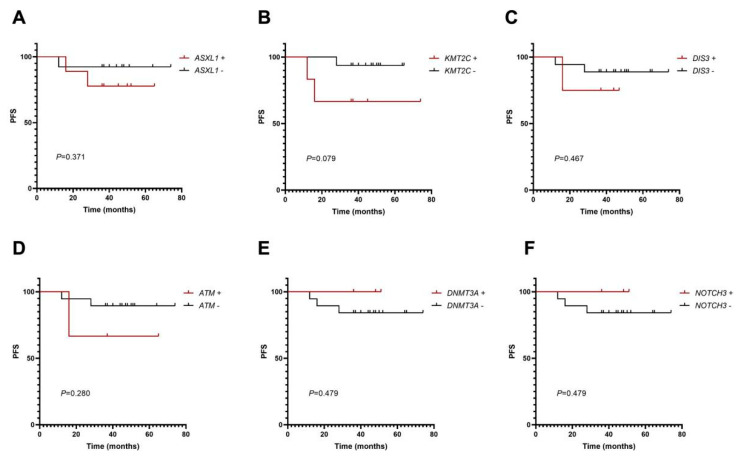
Analysis of *ASXL1* (**A**), *KMT2C* (**B**), *DIS3* (**C**), *ATM* (**D**), *DNMT3A* (**E**) and *NOTCH3* (**F**) mutations on PFS in CML-CP patients.

**Figure 5 cancers-14-05752-f005:**
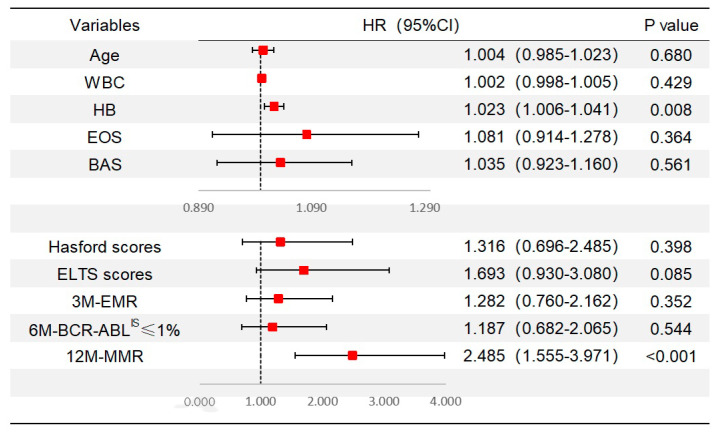
Multivariate analysis on achieving MR^4.5^.

**Table 1 cancers-14-05752-t001:** Baseline characteristics of CML-CP patients and CML-CP patients performed by NGS.

Variables	CML-CP	CML-CP with NGS
Age, years, median (range)	45 (18–86)	50 (25–84)
Sex		
Male (*n*, %)	83 (55.0%)	11 (50.0%)
female (*n*, %)	68 (45.0%)	11 (50.0%)
WBC counts, ×10^9^/L, median (range)	94.8 (2.5–524.5)	59 (11.4–367.0)
HB, g/L, median (range)	112 (43–173)	115 (62–162)
PLT counts, ×10^9^/L, median (range)	583 (14–3526)	529 (110–1558)
Percentage of EOS, %, median (range)	2.4 (0.0–14.0)	2.5 (0–14)
Percentage of BAS, %, median (range)	4.3 (0.0–15.3)	5.4 (2–15.3)
Splenomegaly, cm, median (range)	3.5 (0.0–20.0)	4.7 (0–8.3)
Socal score		
Low risk (*n*, %)	71 (47.0%)	7 (31.8%)
Medium risk (*n*, %)	56 (37.1%)	5 (22.7%)
High risk (*n*, %)	24 (15.9%)	10 (45.5%)
Hasford score		
Low risk (*n*, %)	82 (54.3%)	8 (36.4%)
Medium risk (*n*, %)	54 (35.8%)	10 (45.5%)
High risk (*n*, %)	15 (9.9%)	4 (18.2%)
EUTOS score		
Low risk (*n*, %)	141 (93.4%)	19 (86.4%)
High risk (*n*, %)	10 (6.6%)	3 (13.6%)
ELTS score		
Low risk (*n*, %)	101 (66.9%)	11 (50.0%)
Medium risk (*n*, %)	38 (25.2%)	8 (36.4%)
High risk (*n*, %)	12 (7.9%)	3 (13.6%)
3M-EMR		
Yes (*n*, %)	103 (68.2%)	10 (45.5%)
No (*n*, %)	48 (31.8%)	12 (54.5%)
6M-*BCR∷ABL*^IS^ ≤ 1%		
Yes (*n*, %)	102 (67.5%)	12 (54.5%)
No (*n*, %)	49 (32.5%)	10 (45.5%)
12M-MMR		
Yes (*n*, %)	83 (55.0%)	11 (50.0%)
No (*n*, %)	68 (45.0%)	11 (50.0%)

Abbreviations: WBC, white blood cell; HB, hemoglobin; PLT, platelet; EOS, eosinophil granulocyte; BAS, basophilic granulocyte; 3M-EMR, achieve early molecular response at 3 months; 6M-*BCR∷ABL*^IS^ ≤ 1%, achieve *BCR∷ABL*^IS^ ≤ 1% at 6 months; 12M-MMR, achieve major molecular response at 12 months.

**Table 2 cancers-14-05752-t002:** Mutation analysis in the 12M-MMR group and non 12M-MMR group.

Gene	12M-MMR		*p* Value
	YES (*n* = 11)	NO (*n* = 11)	
*NOTCH3*	0	3	0.214
*RELN*	0	2	0.476
*DIS3*	1	3	0.586
*KMT2C*	4	2	0.635
*ASXL1*	5	4	>0.999
*ATM*	1	2	>0.999
*DNMT3A*	1	2	>0.999
*MED12*	1	1	>0.999
*STAT5A*	1	1	>0.999
*STAG2*	1	1	>0.999
*NOTCH2*	1	0	>0.999
*NOTCH4*	1	0	>0.999
*JAK3*	1	0	>0.999
*PIGA*	1	0	>0.999
*SUZ12*	1	0	>0.999
*ATG2B*	1	0	>0.999
*ABL1*	1	0	>0.999
*CSMD1*	1	0	>0.999
*RUNX1*	1	0	>0.999
*FAT1*	0	1	>0.999
*GNA13*	0	1	>0.999
*TPMT*	0	1	>0.999
*ETV6*	0	1	>0.999
*KMT2A*	0	1	>0.999
*KMT2D*	0	1	>0.999

**Table 3 cancers-14-05752-t003:** Mutation analysis in the 36M-MR^4.0^ group and non 36M-MR^4.0^ group.

Gene	36 Months-MR^4.0^		*p* Value
	YES (*n* = 16)	NO (*n* = 6)	
*ASXL1*	4	5	0.023
*STAT5A*	0	2	0.065
*RELN*	0	2	0.065
*ATM*	1	2	0.169
*NOTCH3*	1	2	0.169
*FAT1*	0	1	0.273
*GNA13*	0	1	0.273
*TPMT*	0	1	0.273
*NOTCH4*	0	1	0.273
*JAK3*	0	1	0.273
*ETV6*	0	1	0.273
*CSMD1*	0	1	0.273
*MED12*	1	1	0.481
*STAG2*	1	1	0.483
*DNMT3A*	3	0	0.532
*KMT2C*	4	2	>0.999
*DIS3*	3	1	>0.999
*KMT2A*	0	1	>0.999
*NOTCH2*	1	0	>0.999
*PIGA*	1	0	>0.999
*SUZ12*	1	0	>0.999
*ATG2B*	1	0	>0.999
*ABL1*	1	0	>0.999
*KMT2D*	1	0	>0.999
*RUNX1*	1	0	>0.999

**Table 4 cancers-14-05752-t004:** Analysis of clinical characteristics, risk stratification and molecular response of 151 CML-CP patients.

Variables	MR4.5		*p* Value
	No (*n* = 26)	Yes (*n* = 125)	
Age, years, median (range)	55 (25–86)	44 (18–84)	0.018
Sex (male/female)	16/10	67/58	0.459
WBC counts, ×10^9^/L, median (range)	120.4 (2.5–366.9)	93.8 (8.5–524.5)	0.165
HB, g/L, median (range)	104 (43–146)	114 (62–173)	0.001
PLT counts, ×10^9^/L, median (range)	602 (14–1409)	579 (100–3526)	0.783
percentage of EOS, %, median (range)	2.8 (0.4–10.0)	2.3 (0.0–14.0)	0.113
percentage of BAS, %, median (range)	3.7 (0.0–8.63)	4.5 (0.0–15.3)	0.056
Splenomegaly, cm, median (range)	5.1 (0.0–14.0)	3.5 (0.0–20.0)	0.432
Socal score (Low/medium/high risk)	9/13/4	62/43/20	0.294
Hasford score (Low/medium/high risk)	10/14/2	72/40/13	0.106
EUTOS score (Low/high risk)	25/1	116/9	0.848
ELTS score (Low/medium/high risk)	13/10/3	88/28/9	0.132
3M-EMR (Yes/No)	11/15	92/33	0.002
6M-*BCR∷ABL*^IS^ ≤ 1% (Yes/No)	13/13	89/36	0.036
12M-MMR (Yes/No)	1/25	82/43	<0.001

Abbreviations: WBC, white blood cell; HB, hemoglobin; PLT, platelet; EOS, eosinophil granulocyte; BAS, basophilic granulocyte; 3M-EMR, achieve early molecular response at 3 months; 6M-*BCR∷ABL*^IS^ ≤ 1%, achieve *BCR∷ABL*^IS^ ≤ 1% at 6 months; 12M-MMR, achieve major molecular response at 12 months.

**Table 5 cancers-14-05752-t005:** Analysis of TKI therapies on the molecular response of 151 CML-CP patients.

Treatment effect	First-Line First-Generation TKI *n* = 115 (A)	First-Line First-Generation Original TKI *n* = 68 (A1)	First-Line First-Generation Generic TKI *n* = 47 (A2)	Second-Line Second-Generation TKI *n* = 27 (B)	First-Line Second-Generation TKI *n* = 9 (C)	*p* Value				
						A vs. C	B vs. C	A1 vs. A2	A1 vs. C	A2 vs. C
3M-EMR (*n*, %)	86 (74.8% )	56 (82.4% )	30 (63.8%r)	9 (33.3%r)	8 (88.9%)	0.584	0.012	0.025	0.985	0.278
6M-*BCR∷ABL*^IS^ ≤ 1% (*n*, %r)	85 (73.9%r)	48 (70.6%r)	37 (78.7%r)	9 (33.3%r)	8 (88.9%)	0.549	0.012	0.329	0.447	0.806
12M-MMR (*n*, %r)	69 (60.0%r)	45 (66.2%r)	24 (48.0%r)	7 (25.9%r)	7 (77.8%)	0.484	0.018	0.104	0.749	0.267
MR4.5 (*n*, %r)	97 (84.3%r)	60 (88.2%r)	37 (78.7%r)	20 (74.1%r)	8 (88.9%)	>0.999	0.643	0.168	>0.999	0.806

Abbreviations: 3M-EMR, early molecular response at 3 months; 6M-*BCR∷ABL*^IS^ ≤ 1%, *BCR∷ABL*^IS^ ≤ 1% at 6 months; 12M-MMR, major molecular response at 12 months.

**Table 6 cancers-14-05752-t006:** Analysis of scoring systems on the molecular response of 151 CML-CP patients.

Scoring System	Risk Stratification	3 Months ≤ 10% (Yes/Nor)	*p* Value	6 Months ≤ 1% (Yes/Nor)	*p* Value	12 Months ≤ 0.1% (Yes/Nor)	*p* Value
EUTOS score	Low risk	99/42	0.103	97/44	0.220	81/60	0.049
	High risk	4/6		5/5		2/8	
Sokal score	Low risk	54/17	0.140	49/22	0.795	41/30	0.777
	Medium risk	35/21		36/20		30/26	
	High risk	14/10		17/7		12/12	
Hasford score	Low risk	62/20	0.088	58/24	0.621	50/32	0.213
	Medium risk	33/21		35/19		27/27	
	High risk	8/7		9/6		6/9	
ELTS score	Low risk	78/23	0.001	74/27	0.090	65/36	0.004
	Medium risk	21/17		22/16		13/25	
	High risk	4/8		6/6		5/7	

**Table 7 cancers-14-05752-t007:** Analysis of scoring systems on the mutations status of 151 CML-CP patients.

	Mutations (A1)	No Mutations (A2)	ASXL1 Mutations (A3)	Other Mutations (A4)	*p* Value			
					A1 vs. A2	A2 vs. A3	A2 vs. A4	A3 vs. A4
EUTOS score (*n*, %)								
Low risk	19 (86.4%)	122 (94.6%)	8 (88.9%)	11 (84.6%)	0.333	>0.999	0.419	>0.999
High risk	3 (13.6%)	7 (5.4%)	1 (11.1%)	2 (15.4%)				
Socal score (*n*, %)								
Low risk	7 (31.8%)	64 (49.6%)	3 (33.3%)	4 (30.8%)	0.001	0.051	0.012	0.992
Medium risk	5 (22.7%)	51 (39.5%)	2 (22.2%)	3 (23.1%)				
High risk	10 (45.5%)	14 (10.9%)	4 (44.4%)	6 (46.2%)				
Hasford score (*n*, %)								
Low risk	8 (36.4%)	74 (57.4%)	4 (44.4%)	4 (30.8%)	0.149	0.470	0.181	0.633
Medium risk	10 (45.5%)	44 (34.1%)	3 (33.3%)	7 (53.8%)				
High risk	4 (18.2%)	11 (8.5%)	2 (22.2%)	2 (15.4%)				
ELTS score (*n*, %)								
Low risk	11 (50.0%)	90 (69.8%)	6 (66.7%)	5 (38.5%)	0.198	0.836	0.062	0.247
Medium risk	8 (36.4%)	30 (23.3%)	3 (33.3%)	5 (38.5%)				
High risk	3 (13.6%)	9 (7.0%)	0 (0.0%)	3 (23.1%)				

## Data Availability

The data presented in this study are available upon request from the corresponding author.

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
