# Peer review of "Targeted Next-Generation Sequencing Identifies Additional Mutations Other than *BCR∷ABL* in Chronic Myeloid Leukemia Patients: A Chinese Monocentric Retrospective Study"

_cancers, 2022, doi:10.3390/cancers14235752_

Round 1

Reviewer 1 Report

This article explores the mutation spectrum other than BCR/ABL1 influencing the response to TKI treatment and prognosis, compares clinical and hematological characteristics in CML-CP patients treated by 1stG-TKI or 2ndG-TKI and validates four scoring systems in the prediction of TKI efficacy. The content of data is abundant, and tables and figures are well designed and presented, therefore, it is easy to understand and would be useful for readers of the manuscript. However, I have some minor comments need the authors to address as following:

1.       Please use the appropriate reference for the most recent NCCN guidelines rather than a 2020 or 2021 reference and update reference in appropriate places accordingly.

2.       I did not see the accurate definition of PFS. Please define progression in the section of “NGS detection and response assessment”, is it accelerated phase or blast phase or both?

3.       In the Methods section, the number of total patients with NGS performed should be detailed.

4.    There is no explanation in the article of the timepoint of NGS testing. This should be clarified.

Author Response

Point 1: Please use the appropriate reference for the most recent NCCN guidelines rather than a 2020 or 2021 reference and update reference in appropriate places accordingly.

Response 1: Thanks for your advice. I have replaced “Deininger, M.W.; Shah, N.P.; Altman, J.K.; Berman, E.; Bhatia, R.; Bhatnagar, B.; DeAngelo, D.J.; Gotlib, J.; Hobbs, G.; Maness, L.; et al. Chronic Myeloid Leukemia, Version 2.2021, NCCN Clinical Practice Guidelines in Oncology. J Natl Compr Canc Netw 2020, 18, 1385-1415, doi:10.6004/jnccn.2020.0047.” by “Gerds, A.T.; Gotlib, J.; Ali, H.; Bose, P.; Dunbar, A.; Elshoury, A.; George, T.I.; Gundabolu, K.; Hexner, E.; Hobbs, G.S.; et al. Myeloproliferative Neoplasms, Version 3.2022, NCCN Clinical Practice Guidelines in Oncology. J Natl Compr Canc Netw 2022, 20, 1033-1062, doi:10.6004/jnccn.2022.0046.”

Point 2: I did not see the accurate definition of PFS. Please define progression in the section of “NGS detection and response assessment”, is it accelerated phase or blast phase or both?

Response 2: Thanks for your advice. Progression was defined as accelerated phase or phase or blast in our manuscript. I have added the description of the definition of “progression” in the Methods section “2.2. NGS detection and response assessment”.

Point 3: In the Methods section, the number of total patients with NGS performed should be detailed.

Response 3: Thanks for your advice. I have added the number of total patients performed by NGS in the Methods section “2.2. NGS detection and response assessment”. 

Point 4: There is no explanation in the article of the timepoint of NGS testing. This should be clarified.

Response 4: Thanks for your advice. I have added the timepoint of NGS testing in the Methods section “2.2. NGS detection and response assessment”. All the 22 CML patients were performed by NGS at diagnosis.

Reviewer 2 Report

Shiwei Hu 1, 3, †, Dan Chen 1, 3, †, Xiaofei Xu 1, 3, Lan Zhang 1, 3, Shengjie Wang 1, 3, Keyi Jin 1, 3, Yan Zheng 1, 3, Xiaoqiong 6 Zhu 1, 3, Jie Jin 2, 3, and Jian Huang 1,2,3,*

Targeted Next-Generation Sequencing Identifies Additional Mutations Other Than BCR ABL in Chronic Myeloid 3 Leukemia Patients: A Chinese Monocentric Retrospective Study

NGS myeloid targeted mutation panel is not routinely performed in patients with CML BCR-ABL1 positive which PCR for ABL1 mutations are widely tested. The clinical roles of somatic gene mutations in CML are largely unknown. The study retrospectively analyzed additional gene mutations in only 22 CML patients in a cohort of 151 patients with limited information in the subgroup.  Increased case number of CML patients with NGS is requested. The results showed a high frequency of ASXL1 mutation in patients who did not achieve molecular remission after TKI therapy, but it is unclear. In addition, the authors have analyzed certain clinical parameters, HgB, and MMR and Sokal, Hasford, EUTOS and ELTS score etc. and determined their roles in response to TKI therapy etc. Revision is recommended before consideration for publication in the peer reviewed journal.

Critiques:

1. The study is composed of two major parts: 1). retrospective study of 151-CML cohort and analyze the risk factors and compare differences on molecular responses among 4 scoring systems etc; 2). Analysis of additional gene mutations on 22 CML patients. The number of patients with available NGS data is too low to generate a big conclusion. In addition, the clinical characteristics of 22 patients were missing. Importantly the patients in the subgroup missed information regarding type of TKI treatment, ABL-1 mutation status, and response to TKIs etc.  The relationship between BCR-ABL1 KD and additional gene mutation e.g. ASXL1 was not illustrated.

                 Please refer to Schoenfeld L Leukemia 2022, PMID 35902731

2). The author adopted 25-gene panel for the study, however, where several key genes e.g., TP53 were not included in the panel. The cases with NGS 36 months follow-ups (36-M-MR4.0) were further subclassified according to pathway. The genes included in each cluster were not documented and number of genes in certain pathway were likely limited. Thus, the results in the Table 3 and 4 might not be useful to predict molecular response.

3). None of Sokol, Hasford, EUTOS and ELTs score system included molecular parameters. Please reanalyzed based on the mutation status.  Non-useful data should be omitted.

Refer to Schoenfeld L Leukemia 2022, PMID 35902731

                e.g., EUTO score                                               mutations            no mutations     ASXL1 mutation

                low

                high

4). The manuscript which is a bit too long and not well focused.

Author Response

Point 1: The study is composed of two major parts: 1). retrospective study of 151-CML cohort and analyze the risk factors and compare differences on molecular responses among 4 scoring systems etc; 2). Analysis of additional gene mutations on 22 CML patients. The number of patients with available NGS data is too low to generate a big conclusion. In addition, the clinical characteristics of 22 patients were missing. Importantly the patients in the subgroup missed information regarding type of TKI treatment, ABL-1 mutation status, and response to TKIs etc.  The relationship between BCR-ABL1 KD and additional gene mutation e.g. ASXL1 was not illustrated.

                 Please refer to Schoenfeld L Leukemia 2022, PMID 35902731

Response 1: Thanks for your advice. We have added the clinical characteristics of 22 patients with NGS in Table 1, including baseline clinical characteristics, four scoring systems and response to TKIs. Since the transcript type of BCR::ABL1 was explored by RNA sequencing, our sequencing was implemented at the DNA level, so we could not provide relevant analysis. We will expand our NGS data and perform on the RNA level to explore further in the near future. 

Point 2: The author adopted 25-gene panel for the study, however, where several key genes e.g., TP53 were not included in the panel. The cases with NGS 36 months follow-ups (36-M-MR4.0) were further subclassified according to pathway. The genes included in each cluster were not documented and number of genes in certain pathway were likely limited. Thus, the results in the Table 3 and 4 might not be useful to predict molecular response.

Response 2: Thanks for your advice. We have added a table to introduce the genes included in each pathway as a supplement at the beginning of the revision. However, considering the advice that several key genes were not included in the panel, the number of genes in certain pathway were likely limited, and the manuscript which is a bit too long and not well focused, we deleted Table 3 and 4 and the text relating to this part to make this manuscript more focused. 

Point 3: None of Sokol, Hasford, EUTOS and ELTs score system included molecular parameters. Please reanalyzed based on the mutation status.  Non-useful data should be omitted.

Refer to Schoenfeld L Leukemia 2022, PMID 35902731

                e.g., EUTO score                 mutations            no mutations     ASXL1 mutation

                       low

                       high

Response 3: Thanks for your advice. We have provided a table displayed the relationship between four scoring systems and the mutation status as advised, which was shown in Table 7. The results showed Socal score could well distinguished mutated subjects and non-mutated subjects, while there was no ideal scoring system in predicting the mutation status especially ASXL1 mutations. This result indicated ASXL1 could serve as an additional prognostic factor and be incorporated into scoring system to better predict the molecular response of CML patients. We have added corresponding analysis in the revised manuscript.

Point 4: The manuscript which is a bit too long and not well focused.

Response 4: Thanks for your advice. We have deleted the graphics and text related to gene functional enrichment, and added relevant analysis of the relationship of four scoring systems and the mutation status to make this manuscript much more focused.
